# Effects of Antithrombotic Agents on Ophthalmological Outcomes, Cardiovascular Risk, and Mortality in Hypertensive Patients with Retinal Vein Occlusion: An Exploratory Retrospective Study

**DOI:** 10.3390/medicina57101017

**Published:** 2021-09-25

**Authors:** Federica Bertoli, Bruno Bais, Daniele De Silvestri, Barbara Mariotti, Daniele Veritti, Alessandro Cavarape, Cristiana Catena, Paolo Lanzetta, Leonardo Alberto Sechi, GianLuca Colussi

**Affiliations:** 1Division of Ophthalmology, Monfalcone-Gorizia Hospital (ASUGI), 34074 Monfalcone, Italy; federica.bertoli8@gmail.com; 2Thrombosis Prevention Unit, Division of Internal Medicine, Academic Hospital of Udine (ASUFC), 33100 Udine, Italy; bruno.bais@asufc.sanita.fvg.it (B.B.); daniele.desilvestri@asufc.sanita.fvg.it (D.D.S.); barbara.mariotti@asufc.sanita.fvg.it (B.M.); 3Department of Medicine-Ophthalmology, University of Udine, 33100 Udine, Italy; daniele.veritti@uniud.it (D.V.); paolo.lanzetta@uniud.it (P.L.); 4Hypertension Unit, Department of Medicine, University of Udine, 33100 Udine, Italy; alessandro.cavarape@uniud.it (A.C.); cristiana.catena@uniud.it (C.C.); leonardo.sechi@uniud.it (L.A.S.)

**Keywords:** visual acuity, cardiovascular disease, survival analysis, antiplatelet drugs, anticoagulants

## Abstract

*Background and objectives*: Because few data are available, the aim of this study is to analyze the effects of antithrombotic agents (ATAs) on visual function and long-term risk of cardiovascular events and mortality in hypertensive patients with retinal vein occlusion (RVO). *Materials and methods*: Hypertensive patients with RVO were consecutively selected from 2008 to 2012 and followed for a median of 8.7 years. Ophthalmologists evaluated and treated RVO complications, and best-corrected visual acuity (BCVA) was checked at each visit during the first one year of follow-up. Survival analysis was conducted on the rate of the composite endpoint of all-cause deaths or non-fatal cardiovascular events. *Results*: Retrospectively, we collected data from 80 patients (age 68 ± 12 years, 39 males). Central and branch RVO was present in 41 and 39 patients, respectively, and 56 patients started ATAs (50 antiplatelet drugs, 6 warfarin, and 2 low-molecular weight heparin). Average BCVA of the cohort did not change significantly during one-year of follow-up. The only predictor of BCVA was the baseline BCVA value. There was a reduction in proportion and severity of macular edema and an increase in the cumulative proportion of retinal vein patency reestablishment during the follow-up, independent of treatment. ATAs had no effects on one-year BCVA, intraocular complications, or the composite endpoint rate. *Conclusions*: In this exploratory study, ATAs had no effect on BCVA during the first one year of follow-up and on the composite endpoint during the long-term follow-up. Further prospective studies need to be conducted with an accurate standardization of the intraocular and antithrombotic treatment to define the positive or negative role of ATAs in hypertensive patients with RVO.

## 1. Background and Objectives

Retinal vein occlusion (RVO) is a common cause of visual loss in older age and in patients with atherosclerotic cardiovascular risk factors [1]. In addition, some observations showed that, in the general population, RVO correlates with an elevated risk of future cardiovascular events such as myocardial infarction and stroke [2]. However, when outcomes were corrected for the cardiovascular risk, some studies failed to observe such an association [3,4]. Patients with RVO have a high prevalence of hypertension, diabetes, hypercholesterolemia, smoking history, and carotid plaque disease, which are “classical” risk factors for major cardiovascular events and mortality [5]. Therefore, we have no clear evidence of whether RVO is just a bystander between “classical” risk factors and cardiovascular event occurrence or if it is causally involved.

Hypertension is the most prevalent risk factor for RVO, which is found in over 60% of patients with an RVO episode [6]. In the Gutenberg study, hypertension and atrial fibrillation were associated with increased probability of branch retinal vein occlusion (BRVO), whereas age and history of stroke were associated with the central form (central retinal vein occlusion CRVO) [7]. Pacella et al. confirmed that hypertension is a powerful predictor of RVO and that hypertension was associated with a worse macular edema and ocular disease severity [8]. Although RVO affects 0.6% of the general population, in hypertensive patients, RVO can be four times more frequent [9]. Therefore, RVO can be considered as a hypertension-related target organ damage [10]. Few data are available about ophthalmological outcomes, cardiovascular risk, and mortality of hypertensive patients with RVO.

Since in the general population RVO is associated with an elevated cardiovascular risk, patients with RVO are treated with antithrombotic agents (ATAs) in primary or secondary cardiovascular prevention. Although antithrombotic therapy, based on anti-platelet drugs or anticoagulants, has been shown to improve visual acuity [11], this therapy is not standardized in patients with RVO without other clinical conditions that require its use [12]. Hence, the systematic impact of antithrombotic therapy on visual acuity, retinal vein patency, adverse retinal vascular complications, and long-term mortality in hypertensive patients with RVO has not been established yet.

The aim of this study was to analyze ophthalmological outcomes, cardiovascular risk, and long-term mortality in a retrospective cohort of hypertensive patients with RVO according to whether or not they started ATAs at RVO diagnosis.

## 2. Materials and Methods

Patients with an RVO episode were consecutively recruited in our Thrombosis Prevention Unit, between November 2008 and April 2012. Patients were addressed to the unit from the Ophthalmology division, where they had been previously evaluated for acute vision loss. We included patients with RVO of 18 years or older, of both sexes, and with a diagnosis of hypertension. We excluded pregnant women, patients with a major cardiovascular event occurred within the previous 6 months, with a glomerular filtration rate lower than 30 mL/min/1.73 m^2^, or those who did not give consent to use personal data. Hypertension history was assessed by clinical records or by finding out the use of antihypertensive drugs.

Blood pressure was measured with an automatic oscillometric sphygmomanometer (HEM-7155-E, Omron Healthcare Europe B.V., Hoofddorp, The Netherlands) after 15 min of lying position on the dominant arm. The average of three measurements taken five minutes apart higher than 140/90 mm Hg was suspected for uncontrolled hypertension. Severely high blood pressure levels (greater than 180/120 mm Hg) were treated with antihypertensive drugs. Antihypertensive therapy was adjusted in those patients with uncontrolled blood pressure levels. The control of blood pressure was confirmed by evaluating the home blood pressure diary 4 to 8 weeks later. Suspected “white coat” or “masked” hypertension were confirmed by ambulatory blood pressure monitoring (TM-2430, A&D Company Limited, Tokyo, Japan), as previously described [13]. The antihypertensive therapy was started or adjusted by adding new or modulating current antihypertensive therapy according to the physician’s decision to maintain blood pressure lower than 140/90 mm Hg. Patients started antithrombotic therapy at the first visit according to clinical conditions and physician preferences. Antithrombotic therapy comprised 100 mg acetylsalicylic acid, 500 mg ticlopidine, and low-molecular weight heparin (LMWH) or warfarin. If warfarin was started, the international normalized ratio (INR) was checked and warfarin dose adjusted as needed to obtain an INR between 2 and 3. LMWH was used at a therapeutic dose adjusted for body weight for at least 90 days. The principal reason for starting an antiplatelet drug or warfarin was the elevated baseline cardiovascular risk or the new diagnosis of atrial fibrillation. LMWH was started because of the physician’s subjective concern of a new RVO episode in the same or fellow eye.

General clinical characteristics, anthropometric variables, and cardiovascular risk factors were collected, and a clinical examination was performed on the first visit. Laboratory tests, including biochemical cardiovascular, pro-inflammatory risk factors, and renal function were prescribed during the first visit, and results were checked with blood pressure re-evaluation at a second visit 4 to 8 weeks later. Plasma total and high-density lipoprotein (HDL) cholesterol, triglycerides, glycated hemoglobin, creatinine levels, fibrinogen, and C-reactive protein (CRP) were measured with standard methods in the centralized laboratory facility of the hospital. Low-density lipoprotein (LDL) cholesterol was calculated with Friedewald’s formula. Body mass index was calculated as weight in kilograms over the squared height in meters. Glomerular filtration rate was estimated with the Modification of Diet in Renal Disease (MDRD) study equation corrected for the body surface area of 1.73 m^2^. Age-weighted Charlson Comorbidity Index (CCI) was calculated to account for the comorbidities burden by adding 1 point to the original CCI score each decade above 40 years of age [14]. We defined a history of cardiovascular disease as the occurrence of stroke, transitory ischemic attack, coronary artery disease, peripheral artery disease, myocardial infarction, atrial fibrillation, or heart failure 6 months before RVO diagnosis. Fatal and non-fatal cardiovascular events and all-cause deaths were assessed by checking clinical records and by telephone contact when needed. Non-fatal cardiovascular events were defined as the occurrence of a non-fatal myocardial infarction, stroke, or transitory ischemic attack. The last month of patients’ cohort assessment was December 2019.

All patients agreed to take part in the study by telephone contact, and informed consent was collected when possible. All the procedures were under the Helsinki Declaration of 1975 (as revised in 1983). The Institutional Review Board of the University of Udine approved the study protocol on 20 July 2021 (protocol number: 051/2021).

### 2.1. Ophthalmological Evaluation

Patients with sudden vision loss or blurred vision were seen at the Ophthalmology division of the Academic Hospital of Udine. RVO was diagnosed at first ophthalmological visit by slit-lamp dilated ophthalmic fundus examination and subsequently confirmed by fluorescein angiography, as previously described [15]. Spectral domain or swept-source optical coherence tomography (OCT) was performed to quantify retinal thickening from macular fluid accumulation (macular edema) at the physician’s discretion. RVO was classified in CRVO or BRVO and ischemic or non-ischemic forms. The few cases of hemicentral RVO were included in CRVO for statistical purposes. Patients with signs of significant retinal ischemia, neovascularization, or with macular edema were treated with laser photocoagulation therapy, intravitreal corticosteroid implant, or intravitreal vascular endothelial growth factor inhibitors (anti-VEGF) at physicians’ discretions. The other patients were observed over the follow-up time without treatment unless signs of macular edema or neovascularization/ischemia appeared. At each control, best-corrected visual acuity (BCVA) was assessed with the Early Treatment Diabetic Retinopathy Study (ETDRS) chart and expressed as the logarithm of the minimum angle of resolution (logMAR) units, as previously described [16]. Patients with RVO were followed in the Ophthalmology division at 1, 3, 6, and 12 months from RVO diagnosis. As shown in previous studies [11], six months to one year of follow-up suffices to observe ophthalmological outcomes associated with the use of ATAs after an acute RVO episode. Ophthalmological outcomes comprised determining at each visit BCVA, macular edema, vitreous hemorrhages, neovascularization of the iris (rubeosis iridis) and retina, neovascular glaucoma, and retinal vein patency. Macular edema was classified according to the central retinal thickness (CRT) measures at OCT as absent (CRT < 250 µm), mild (CRT between 250 and 299 µm), moderate (CRT between 300 µm and 400 µm), or severe (CRT > 400 µm), scoring 0, 1, 2, or 3, respectively, by the same ophthalmologist [17].

### 2.2. Statistical Analysis

Data are presented as mean ± standard deviation for normally distributed variables or as median and interquartile range (IQR) for the skewed ones. Means comparison was performed with Student’s t test for the normally distributed variables and with the non-parametric Wilcoxon test for the non-normally distributed variables. Proportion comparison was performed with Fisher’s exact test and, for trend proportions across time, with the chi-square test for trends. The longitudinal relationship between continuous dependent variables and independent predictors over time was analyzed with the linear mixed-effect model for repeated variables. Missing values in longitudinal analysis were imputed with the last observation carried forward method. Survival analysis was conducted on the composite endpoint made up of all-cause deaths or major non-fatal cardiovascular events. Predictors of the composite endpoint were determined by univariate analysis based on the Cox proportional hazards model. Multivariate models were built by including all statistically significant predictors in the univariate analysis. Results of the survival analysis were presented as hazard ratio (HR) and the 95% confidence interval (CI). Survival probability for the composite endpoint according to antithrombotic treatment was presented as Kaplan-Meier curves and 95% CI. Statistics were performed with the log-rank test. Because of the exploratory nature of the study, we did not previously define the sample size. We considered significant for excluding the null hypothesis a probability (*p*) lower than 5%. Statistical analysis was performed with the free R software, version 4.1.0 [18].

## 3. Results

For this study, we recruited 80 patients with complete data. Fifty-six (70%) started treatment with ATAs after RVO diagnosis. Of these patients, 40 started acetyl salicylic acid, 10 ticlopidine, 6 warfarin, and 2 LMWH. Treated patients had a higher prevalence of cardiovascular disease history than those not treated. There were no differences regarding other general clinical characteristics, laboratory variables, and baseline ophthalmological outcomes between treated and non-treated patients (Table 1).

### 3.1. Ophthalmological Outcomes

Patients presented only one eye affected by RVO, and no new RVO events were documented in the same or in the fellow eye during one year of follow-up. CRVO was diagnosed in 41 eyes, 2 of which were with the hemicentral form. The remaining eyes showed the BRVO form. Ischemic presentation of CRVO and BRVO occurred in 4 and 12 eyes, respectively. Forty-one eyes received an ophthalmological treatment, while the others were only observed during follow-up. Of the treated eyes, 31 received laser therapy because of retinal ischemia/neovascularization (18 eyes) or macular edema (13 eyes), 16 received intravitreal anti-VEGF, and 3 received intravitreal steroids. At baseline, there were no differences in ophthalmological variables between patients treated or not with ATAs (Table 1). Average BCVA did not change significantly during the one-year follow-up (−0.005 ± 0.045 logMAR, *p* = 0.298), and there were no difference between patients treated or not with ATAs (treated: −0.007 ± 0.055 logMAR, *p* = 0.235; not treated: 0.0003 ± 0.079 logMAR, *p* = 0.972, Figure 1). The only predictor of BCVA during the one-year follow-up was the baseline BCVA value (+0.09 logMAR each 0.1 logMAR of baseline BCVA, *p* < 0.001). At the end of one-year follow-up, there were no differences in the proportions of ophthalmological outcomes between patients treated or not with ATAs (Table 2). The proportion and severity of macular edema decreased, and the cumulative proportion of retinal vein patency reestablishment increased across follow-up time independently of treatment (Figure 2).

### 3.2. Survival Analysis

The cohort of patients was followed for a median time of 8.7 [IQR 7.5–9.9] years. At the end of the survey, 29 patients presented the composite endpoint, 23 died, and 6 had recognized fatal and 7 non-fatal cardiovascular events. After RVO diagnosis, the median time of composite endpoint occurrence was 5.3 [IQR 3.9–7.9] years; median survival time of dead patients was 4.8 [IQR 2.9–7.1] years. At the end of follow-up, there was no difference in the prevalence of the composite endpoint and its components and in median survival time between patients treated and not treated with ATAs (Table 2). The composite endpoint rate did not differ between patients treated or not with ATAs during long-term follow-up (Figure 3). Direct predictors of the composite endpoint were age, CCI, prevalence of cardiovascular disease, and levels of plasma CRP, whereas the only inverse predictor was the estimated glomerular filtration rate (Table 3). Only age and CCI remained independent predictors in multivariate analysis (Table 3).

## 4. Discussion

In this study, the proportion and severity of macular edema decreased, and the cumulative proportion of retinal vein patency reestablishment increased over time. Conversely, BCVA did not improve, and ATAs use was not associated with changes in BCVA or intraocular complications during the one-year follow-up. In addition, ATA use was not associated with a difference in the combined endpoint rate in long-term follow-up.

Hypertension is associated with RVO, especially with the BRVO form [5,19]. This association is stronger in uncontrolled hypertension, whereas controlled hypertensive patients have a corrected RVO risk that is equivalent to that of the normotensive population [9]. Predisposing factors for RVO are the presence and severity of retinal arteriovenous nicking and vessel tortuosity that are characteristic of hypertension-related retinopathy and are more severe in uncontrolled hypertension [12,20]. These anatomical vascular changes induce a turbulent hematic flow that promotes thrombus formation [21]. In addition, hypertension is associated with a pro-thrombotic state [22,23]. Since RVO occurs when hemostasis is impaired according to the Virchow’s triad (endothelial dysfunction, hemodynamic vascular changes, and blood hyperviscosity) [24], ATAs have been considered as a rational medical treatment [25]. However, treatment of RVO with ATAs remains controversial because of conflicting results of previous studies. To note, the indication for using ATAs in RVO is driven mainly by the underlying cardiovascular conditions [26]. In our study, treated patients had a higher prevalence of cardiovascular disease, which was the principal reason for the ATA treatment.

In a systematic review, Squizzato et al. analyzed 384 cases of RVO in six randomized controlled trials [11]. Patients were treated with either acetyl salicylic acid, warfarin, ticlopidine, fibrinolytic therapy, hemodilution, or placebo. A partial improvement of visual acuity was observed with each intervention, especially with LMWH [11]. In our study, visual acuity did not improve significantly, probably because of the opposite effects of the slight improvement with ATAs and the tendency toward worsening of untreated eyes (Figure 1). In addition, in our study, most patients were started on antiplatelet drugs (acetylsalicylic acid or ticlopidine) instead of LMWH, and some authors observed that heparins can be more effective than antiplatelet drugs for improving visual function in RVO [27]. However, other studies failed to show the superiority of LMWH over acetyl salicylic acid [28]. This point remains controversial.

In this study, we did not observe any association between antithrombotic treatment and improvement of ophthalmological outcomes. Macular edema is the consequence of a structural alteration of the macular microvasculature secondary to retinal ischemia that contributes to impaired visual acuity [29]. Although there is evidence of the improvement of visual acuity with ATAs, the reason for such an improvement is not clear. To our knowledge, no studies have reported the effects of ATAs on macular edema in RVO. Therefore, since ATAs should speed up thrombus dissolution, it seems unlikely that RVO is the simple consequence of an intravascular thrombogenic process [30]. LMWH, which appears to be the most effective agent in RVO, reduces inflammation and promotes vasodilation beyond its anticoagulant properties [31,32]. Other pathophysiological mechanisms for RVO should be considered and this important point further studied.

The principal adverse effect of ATAs is bleeding. Using these drugs in thromboembolic diseases or in systemic thrombosis is justified because the clinical benefits outweigh the risks. However, in RVO, the balance between clinical benefits and risks remained controversial. ATAs have been associated with increased ophthalmological adverse outcomes in one large retrospective study [33]. In patients with central and hemicentral RVO, Hayren et al. observed a greater severity of retinal hemorrhages in those taking acetylsalicylic acid or warfarin. In addition, the use of acetylsalicylic acid or warfarin was associated with a deterioration in visual acuity [33]. In our study, we did not observe any increase in adverse intraocular hemorrhages in patients taking ATAs, and most of our patients were taking acetylsalicylic acid or warfarin, as in the Hayren’s study. However, it should be considered that adverse ocular events might have been masked by the type and timing of the intraocular treatment, since intraocular treatment could have limited retinal bleeding problems. It is uncertain whether antiplatelet drugs or oral anticoagulants could be deleterious in RVO, but it has been shown that LMWH is associated with a 78% risk reduction of adverse retinal events compared to acetylsalicylic acid [34]. Further studies should assess prospectively the effectiveness and safety of the antithrombotic therapy in RVO, after an accurate standardization of the ophthalmological and antithrombotic treatment.

Several studies have reported an elevated risk of stroke, myocardial infarction, heart failure, peripheral arterial disease, and all-cause mortality in patients with RVO compared with controls without RVO [35] with some discordant results between the CRVO and BRVO form. CRVO has been associated with increased mortality [4], whereas in BRVO this association seems weaker [3,36]. In our study, we did not observe a difference in the composite endpoint of all-cause deaths or non-fatal cardiovascular events in RVO and between CRVO and BRVO forms. Independent predictors of mortality and cardiovascular events in our study were age and the comorbidity burden, and both these predictors are common risk factors for RVO [37]. It remains uncertain whether RVO by itself can be a major cardiovascular risk factor or just a bystander and whether CRVO or BRVO, as the potential expression of two separate diseases, can have different prognostic roles in cardiovascular prevention. This point should be further analyzed.

This study has several important limits to discuss. First, because we included consecutive patients with completed baseline and follow-up data, it resulted in a small sample size, and no power calculation was performed. Because of this important limit, this study is to be considered just exploratory. Second, ophthalmological and antithrombotic treatments were heterogeneous. The type of intraocular treatment and different use of ATAs could have influenced the effect of ATAs on ophthalmological outcomes and on the adverse ocular consequences. This should be considered with attention because the effect of ATAs in RVO pathology might be minimal and masked by uncontrolled or unstandardized confounding factors. Third, treatment assignment was not randomized, but it was guided only by a subjective physician indication based on the cardiovascular risk and concerns about RVO recurrence. This may have added another confounding source for interpreting the results. Fourth, in this study, we used baseline variables to predict the composite endpoint of all-cause mortality or non-fatal cardiovascular events. However, in the long-term follow-up, therapy changes and clinical modifications that have occurred after the initiation of antithrombotic treatment could have changed the natural history of the disease and influenced the study results. In particular, we were confident that continuous use of ATAs and blood pressure levels were controlled during the 1-year ophthalmological follow-up. Conversely, we did not have information about the continuation of treatment and cardiovascular risk factors control during the long-term follow-up.

In conclusion, this exploratory retrospective study shows that, in hypertensive patients with RVO, use of ATAs was neither associated with visual function improvement nor with adverse intraocular complications during one year of follow-up. ATA use in these patients did not change the rate of the composite endpoint of all-cause mortality or non-fatal cardiovascular events in the long-term follow-up. However, because of the important limits of this study, these results need to be confirmed in further prospective studies.

## Figures and Tables

**Figure 1 medicina-57-01017-f001:**
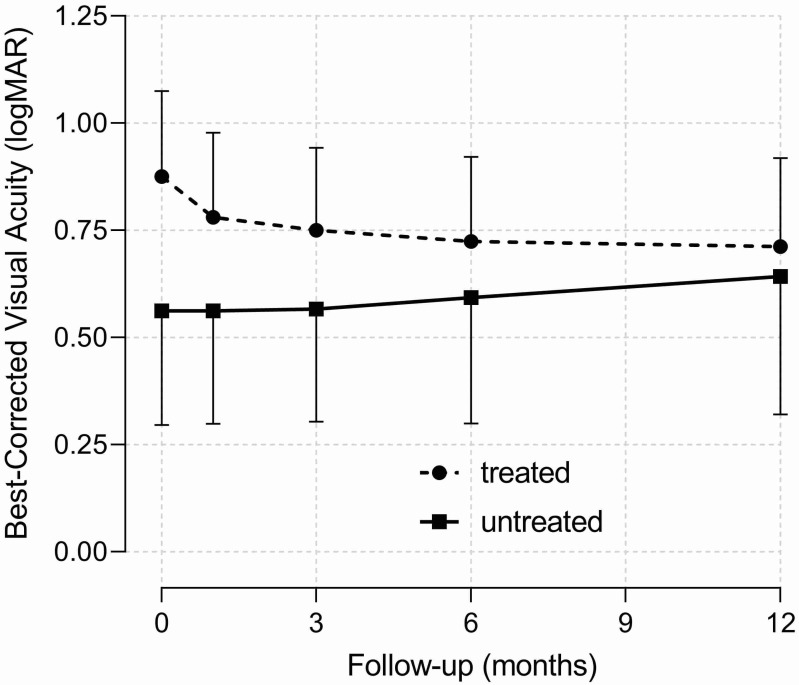
Best-corrected visual acuity of hypertensive patients with retinal vein occlusion at baseline (time 0) and during one-year follow-up after receiving (dashed line) or not (solid line) antithrombotic treatment and respective 95% confidence intervals (vertical capped lines).

**Figure 2 medicina-57-01017-f002:**
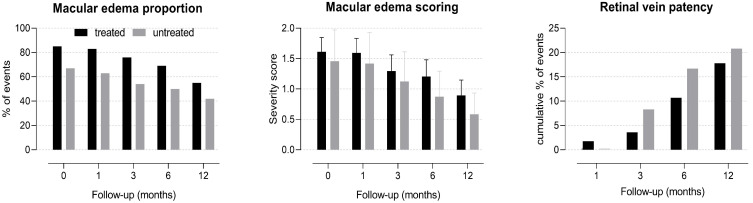
Ophthalmological outcomes of hypertensive patients with retinal vein occlusion during one-year follow-up after receiving (black bars) or not (grey bars) antithrombotic treatment and respective 95% confidence interval (vertical capped lines). *p* for trend over time <0.050 for all series independently of antithrombotic treatment.

**Figure 3 medicina-57-01017-f003:**
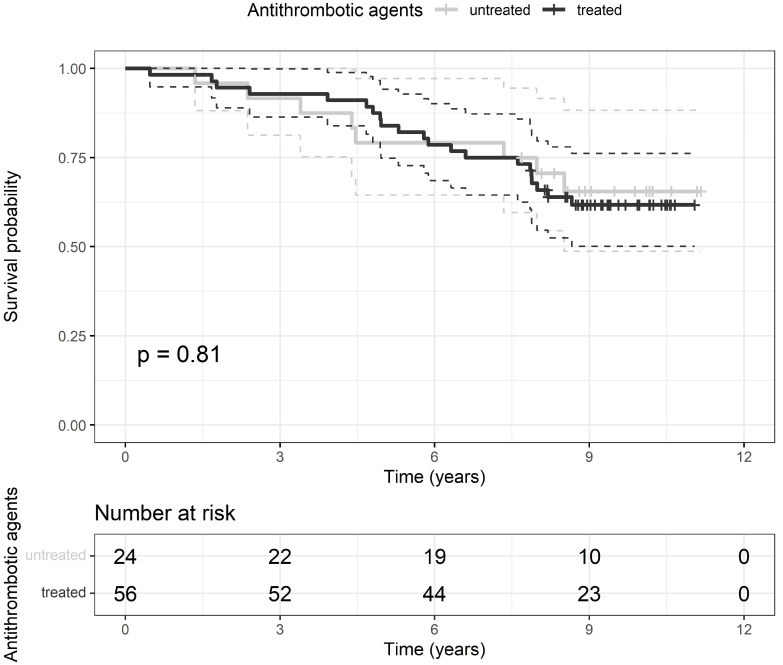
Kaplan-Meier curves of the composite endpoint rate after receiving (black line) or not (grey line) the antithrombotic treatment with the respective 95% confidence interval (dashed black or grey lines). Vertical marks represent censored patients. Below the graph is reported a table with the number of patients at risk treated and untreated with antithrombotic agents at each time-point. *p*, probability for the difference between groups by log-rank test.

**Table 1 medicina-57-01017-t001:** General clinical characteristics, laboratory findings, and ophthalmological variables at RVO presentation of patients treated and untreated with antithrombotic agents.

Baseline Variable	All Patients	Treated	Untreated	*p*
	N = 80	56	24	
Age (years)	68 ± 12	69 ± 12	66 ± 14	0.311
Male sex (*n* (%))	39 (49)	26 (46)	13 (54)	0.696
Charlson Comorbidity Index	3.2 ± 1.9	3.4 ± 1.9	2.6 ± 2.0	0.092
BMI (Kg/m^2^)	26.4 ± 4.8	26.4 ± 4.8	26.6 ± 4.7	0.846
Active smoking (*n* (%))	13 (16)	8 (15)	5 (21)	0.534
CVD (*n* (%))	20 (25)	18 (32)	2 (8)	0.026
Office SBP (mm Hg)	146 ± 19	144 ± 12	149 ± 19	0.359
Office DBP (mm Hg)	82 ± 9	82 ± 8	85 ± 10	0.208
Dyslipidemia (*n* (%))	29 (36)	22 (39)	7 (29)	0.454
Total cholesterol (mg/dL)	218 ± 41	217 ± 40	220 ± 44	0.773
LDL cholesterol (mg/dL)	136 ± 36	135 ± 37	138 ± 33	0.654
HDL cholesterol (mg/dL)	59 ± 16	58 ± 17	61 ± 14	0.452
Triglycerides (mg/dL)	121 ± 61	124 ± 65	115 ± 48	0.497
Diabetes (*n* (%))	14 (18)	11 (20)	3 (12)	0.536
HbA1c (%)	5.7 ± 0.8	5.8 ± 0.8	5.5 ± 0.6	0.065
Plasma fibrinogen (mg/dL)	398 ± 111	399 ± 96	397 ± 142	0.962
CRP (mg/L)	1.93 (0.99–3.87)	2.04 (0.96–3.95)	1.58 (1.01–3.07)	0.712
Plasma creatinine (mg/dL)	1.04 ± 0.40	1.07 ± 0.46	0.98 ± 0.16	0.218
eGFR (mL/min/1.73m^2^)	66 ± 14	64 ± 15	69 ± 12	0.132
*Ophthalmological variables*				
CRVO (*n* (%))	41 (51)	30 (54)	11 (46)	0.628
BRVO (*n* (%))	39 (49)	26 (46)	13 (54)	0.628
BCVA (logMAR)	0.78 ± 0.70	0.85 ± 0.73	0.64 ± 0.62	0.189
Retinal ischemia (*n* (%))	9 (11)	7 (13)	2 (9)	0.905
Retinal neovascularization (*n* (%))	2 (3)	2 (4)	0	-
Hemovitreus (*n* (%))	4 (5)	1 (2)	3 (8)	0.087
Rubeosis iridis (*n* (%))	0	0	0	-
Neovascular glaucoma (*n* (%))	0	0	0	-

RVO, retinal vein occlusion; CVD, history of cardiovascular disease; BMI, body mass index; SBP, systolic blood pressure; DBP, diastolic blood pressure; LDL, low-density lipoproteins; HDL, high-density lipoproteins; HbA1c, glycated hemoglobin; CRP, C-reactive protein; eGFR, estimated glomerular filtration rate; CRVO, central retinal vein occlusion; BRVO, branch retinal vein occlusion; BCVA, best corrected visual acuity; *p*, probability.

**Table 2 medicina-57-01017-t002:** Ophthalmological outcomes, composite endpoint components, and median survival time at the end of follow-up of patients treated and untreated with antithrombotic agents.

Baseline Variable	All Patients	Treated	Untreated	*p*
*Ophthalmological variables*				
Retinal ischemia (*n* (%))	4 (5)	4 (7)	0	-
Retinal neovascularization (*n* (%))	2 (2.5)	2 (3.6)	0	-
Hemovitreus (*n* (%))	1 (1.3)	1 (1.8)	0	-
Rubeosis iridis (*n* (%))	2 (2.5)	2 (3.6)	0	-
Neovascular glaucoma (*n* (%))	3 (3.8)	2 (3.6)	1 (4.2)	0.845
*Composite endpoint components*				
Composite endpoint (*n* (%))	29 (36)	21 (38)	8 (33)	0.803
Median time of composite endpoint occurrence (years)	5.3 (3.9–7.9)	5.8 (4.7–7.9)	4.4 (3.1–7.5)	0.518
All-cause deaths (*n* (%))	23 (29)	16 (29)	7 (29)	0.942
Major fatal cardiovascular events (*n* (%))	6 (7.5)	4 (7.1)	2 (8.3)	0.904
Major non-fatal cardiovascular events (*n* (%))	7 (8.8)	6 (11)	1 (4.2)	0.668
Median follow-up time (years)	8.7 (7.5–9.9)	8.7 (7.4–9.6)	8.7 (7.6–10)	0.797
Median survival time of died patients (years)	4.8 (2.9–7.1)	4.9 (3.5–6.9)	4.4 (2.9–6.2)	0.579

*p*, probability.

**Table 3 medicina-57-01017-t003:** Variables associated with composite endpoint by univariate and multivariate Cox regression analysis.

Baseline Variable	Univariate Analysis			Multivariate Analysis		
	HR	95% CI	*p*	HR	95% CI	*p*
Age (years)	1.153	1.091–1.218	<0.001	1.082	1.008–1.163	0.030
Male sex (yes/no)	1.389	0.668–2.887	0.380			
Charlson Comorbidity Index	2.093	1.620–2.704	<0.001	1.700	1.171–2.471	0.005
Diabetes (yes/no)	0.920	0.351–2.412	0.865			
Dyslipidemia (yes/no)	0.577	0.255–1.303	0.186			
CVD (yes/no)	3.174	1.523–6.614	0.002	0.855	0.338–2.165	0.741
BMI (Kg/m^2^)	0.960	0.887–1.039	0.315			
Active smoker (yes/no)	0.693	0.241–1.993	0.497			
SBP (mm Hg)	1.003	0.984–1.023	0.736			
DBP (mm Hg)	0.999	0.958–1.041	0.949			
Total cholesterol (mg/dL)	0.999	0.990–1.008	0.823			
LDL cholesterol (mg/dL)	1.001	0.990–1.011	0.899			
HDL cholesterol (mg/dL)	1.006	0.984–1.028	0.610			
Triglycerides (mg/dL)	0.994	0.985–1.003	0.174			
HbA1c (%)	1.284	0.860–1.918	0.222			
Fibrinogen (mg/dL)	1.001	0.998–1.004	0.502			
Log (CRP (mg/L))	1.417	1.074–1.868	0.014	1.147	0.840–1.567	0.389
Plasma creatinine (mg/dL)	1.167	0.578–2.357	0.667			
eGFR (ml/min/1.73 m^2^)	0.978	0.957–0.999	0.046	1.003	0.980–1.026	0.820
CRVO (yes/no)	1.033	0.498–2.140	0.931			
BRVO (yes/no)	0.968	0.467–2.006	0.931			
Antithrombotic agents (yes/no)	1.106	0.490–2.498	0.808	0.585	0.218–1.566	0.286

CVD, history of cardiovascular disease; BMI, body mass index; SBP, systolic blood pressure; DBP, diastolic blood pressure; LDL, low-density lipoproteins; HDL, high-density lipoproteins; HbA1c, glycated hemoglobin; CRP, C-reactive protein; eGFR, estimated glomerular filtration rate; CRVO, central retinal vein occlusion; BRVO, branch retinal vein occlusion; HR, hazards ratio; CI, confidence interval; *p*, probability.

## Data Availability

All data are property of the Academic Hospital of Udine and available from the corresponding author upon reasonable request and by institutional approval.

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
