# Peer review of "Effects of Antithrombotic Agents on Ophthalmological Outcomes, Cardiovascular Risk, and Mortality in Hypertensive Patients with Retinal Vein Occlusion: An Exploratory Retrospective Study"

_medicina, 2021, doi:10.3390/medicina57101017_

Round 1
Reviewer 1 Report
For this study, the authors recruited 80 patients with complete data. Fifty-six (70%) started 169treatment with ATAs after RVO diagnosis. Of these patients, 40 started acetyl salicylic 170 acid, 10 ticlopidine, 6 warfarin, and 2 LMWH.
The criteria for selecting an anticoagulant or antiplatelet and the duration of therapy should be outlined, in order to provide clues for data interpretation. Was this treatment prescribed for RVO or for concurrent vascular reasons? This is important
The duration of treatment, for a short period or at long term is crucial for interpreting the outcome. It should be reported and commented.
What about antihypertensive treatment? Was it at target?
Author Response
Reviewer 1
We thanks the reviewer for his/her thoughtful comments. Here, we present our point-by-point responses to the raised concerns. Changes to the original manuscript are highlighted in red in the revised form.
R1: For this study, the authors recruited 80 patients with complete data. Fifty-six (70%) started 169treatment with ATAs after RVO diagnosis. Of these patients, 40 started acetyl salicylic 170 acid, 10 ticlopidine, 6 warfarin, and 2 LMWH. The criteria for selecting an anticoagulant or antiplatelet and the duration of therapy should be outlined, in order to provide clues for data interpretation. Was this treatment prescribed for RVO or for concurrent vascular reasons? This is important.
AU: We totally agree with the reviewer's comments about the importance of reasons for starting antithrombotic agent and prescribing LMWH instead of warfarin or antiplatelet drugs, and about the timing of treatment. However, because of the long retrospective period of the study and the lack of clear indications on antithrombotic treatment in RVO, this information was not completely available. The principal reason for starting antiplatelet drugs or warfarin was the elevated cardiovascular risk at baseline (see Table 1) and new diagnosis of atrial fibrillation, whereas LMWH was started because of the subjective physician's concern of a new RVO event in the fellow eye. We have made a comment on this point and remarked on the limits of the study (page 3, lines 98-101; page 10-11, lines 334-337).
R1: The duration of treatment, for a short period or at long term, is crucial for interpreting the outcome. It should be reported and commented.
AU: We were confident that antiplatelet and warfarin were taken for at least one-year and LMWH for at least 90 days during the ophthalmologic follow-up. Conversely, we did not have information about the continuation of treatment in the log-term follow-up. This has been remarked on the limits of the study (page 11, lines 337-345).
R1: What about antihypertensive treatment? Was it at target?
AU: As for the use of antithrombotic agents, we were confident that blood pressure has been treated and controlled during the 1-year ophthalmologic follow-up (page 3, lines 91-92). However, information about blood pressure control during the long-term follow-up was not available. This point has been added as a limit of the study (page 11, lines 341-345
Reviewer 2 Report
The subject of the paper is very interesting. Despite the advances in ophthalmology, there is still a reserved visual prognosis for the patients affected by retinal vein thrombosis. The fact that the patients were followed-up for a long period of time (mean of 8.7 years) is a strong point of the research.
However, the main reason for prescribing antithrombotic therapy in patients diagnosed with central retinal vein obstruction or branch retinal vein obstruction is to prevent the same event in the fellow eye, not really to improve vision in the affected eye.
It will be interesting for the reader to provide comparative data regarding the incidence of retinal vein thrombosis in the fellow eye between the 2 study groups during the follow-up period.
Also, a STROBE cohort study checklist should be provided as supplementary material
Author Response
Reviewer 2
We thanks the reviewer for his/her thoughtful comments. Here, we present our point-by-point responses to the raised concerns. Changes to the original manuscript are highlighted in red in the revised form.
R2: The subject of the paper is very interesting. Despite the advances in ophthalmology, there is still a reserved visual prognosis for the patients affected by retinal vein thrombosis. The fact that the patients were followed-up for a long period of time (mean of 8.7 years) is a strong point of the research. However, the main reason for prescribing antithrombotic therapy in patients diagnosed with central retinal vein obstruction or branch retinal vein obstruction is to prevent the same event in the fellow eye, not really to improve vision in the affected eye.
AU: We agree with the reviewer's comment and this point has been emphasized in the revised manuscript (page 3, lines 98-101)
R2: It will be interesting for the reader to provide comparative data regarding the incidence of retinal vein thrombosis in the fellow eye between the 2 study groups during the follow-up period.
AU: During the 1-year ophthalmologic follow-up, no new RVO events were documented in the same or in the fellow eye. We have remarked on this information in the revised manuscript (page 5, line 187)
R2: Also, a STROBE cohort study checklist should be provided as supplementary material
AU: We have provided a STROBE checklist as supplemental material
Round 2
Reviewer 2 Report
Some of the issues were addressed in the revised version, but there is still a need for improvement. The methodology of research is not clearly presented. In the abstract (lines 16-17), the authors state that: " Hypertensive patients with RVO were consecutively selected from 2008 to 2012 and followed for a median of 8.7 years, " I found it one of the strengths of the research. However, in the main article, it appears that patients were followed up only up to 12 months from the ophthalmological point of view (lines 198-190). This time is too short to draw any conclusions. Moreover, the antithrombotic treatment is heterogeneous.
Author Response
Reply to Reviewer 2
R2: Some of the issues were addressed in the revised version, but there is still a need for improvement. The methodology of research is not clearly presented. In the abstract (lines 16-17), the authors state that: " Hypertensive patients with RVO were consecutively selected from 2008 to 2012 and followed for a median of 8.7 years, " I found it one of the strengths of the research. However, in the main article, it appears that patients were followed up only up to 12 months from the ophthalmological point of view (lines 198-190). Moreover, the antithrombotic treatment is heterogeneous.
AU: We confirm that the period of the ophthalmological observation and the occurrence of ophthalmological outcome lasted the first year of the follow-up, whereas other endpoints such as cardiovascular events and all-cause mortality were evaluated in the long-term period. We have clarified this point in the abstract (lines 19, 29-30).
R2: This time is too short to draw any conclusions.
AU: As shown in previous studies (see systematic review of Squizzato et al. Ref. 11), six months to one year of follow-up suffices to observe BCVA variations, RVO recurrence, and potential ophthalmological adverse effects associated with the use of antithrombotic agents after an acute RVO episode. We have specified this point in methods (lines 145-147).
R2: Moreover, the antithrombotic treatment is heterogeneous.
AU: We agree with the reviewer and we have highlighted this important point as a limit of the study (lines 335-336).